



Foreign emissions exacerbate PM$_{2.5}$ pollution in China through nitrate chemistry
Jun-Wei Xu[1], Jintai Lin[1]*, Gan Luo[2], Jamiu Adeniran[1], Hao Kong[1]
[1]Laboratory for Climate and Ocean–Atmosphere Studies, Department of Atmospheric and
Oceanic Sciences, School of Physics, Peking University, Beijing, China
[2]Atmospheric Sciences Research Center, University at Albany, Albany, NY, USA
Correspondence: Jintai Lin (linjt@pku.edu.cn).
**Abstract**
Fine particulate matter (PM$_{2.5}$) pollution is a severe problem in China. Research on the sources
of Chinese PM$_{2.5}$ pollution has focused on the contributions of China's domestic emissions.
However, the impact of foreign anthropogenic emissions has typically been simplified or
neglected, partly due to the perception that the short lifetime of PM$_{2.5}$ (a few days) does not
allow long-distance transport. Here we explore the role of foreign anthropogenic emissions in
Chinese PM$_{2.5}$ pollution in 2015 using the GEOS-Chem chemical transport model. We validate
the model simulations with a comprehensive set of observations of PM$_{2.5}$ and its compositions,
including sulfate, nitrate, ammonium, black carbon and primary organic aerosols, over China and
its surrounding regions. We find that 8% of PM$_{2.5}$ (5 µg m$^{-3}$) and 19% of nitrate (2.6 µg m$^{-3}$) over
eastern China in 2015 was contributed by foreign anthropogenic emissions. The contributions
were the highest in January (6.9 µg m$^{-3}$ PM$_{2.5}$, with 68% nitrate) and the lowest in July (2.7 µg
m$^{-3}$ PM$_{2.5}$, with 11% nitrate). Yet, only 30% of such foreign contributions in January was
through direct atmospheric transport. The majority (70%) was instead through chemical
interactions between foreign-transported aerosol precursors and China's domestic emissions of
pollutants. Specifically, the transport of non-methane volatile organic compounds (NMVOCs)
from foreign countries enhanced the atmospheric oxidizing capacity and facilitated the oxidation
of Chinese nitrogen oxides (NO$_x$) to form nitric acid (HNO$_3$) over the eastern China. The
abundance of Chinese ammonia (NH$_3$) further partitioned nearly all HNO$_3$ gas to particulate
nitrate, leading to the considerable foreign contributions of nitrate and PM$_{2.5}$ to the eastern
China. Over southwestern China, foreign anthropogenic emissions contributed 4.9 µg m$^{-3}$ PM$_{2.5}$
concentrations (18% of total PM$_{2.5}$ mass) to Yunnan province, with 37% as organics and 27% as
sulfate. Our findings suggest that foreign anthropogenic emissions play an important role in
Chinese PM$_{2.5}$ pollution, because of direct aerosol transport and, more importantly, chemical
interactions between transported pollutants and China's local emissions. Thus, foreign emission
reductions will be very beneficial for improving Chinese air quality.




**1. Introduction**
China has been severely affected by fine particulate matter ($PM_{2.5}$, particulate matter smaller
than 2.5 µm in aerodynamic diameter) pollution over the past decades from processes of
industrialization and urbanization (Geng et al., 2021; West et al., 2016). Over 1 million
premature deaths associated with $PM_{2.5}$ pollution occur in China every year (Cohen et al., 2017;
Yue et al., 2020; Zhang et al., 2017). In response, the Chinese government imposed stringent
emission controls on primary particles and precursor gases in the 5-year Clean Air Action in
2013 (China State Council, 2013), leading to a nationwide emission reduction of 59% for sulfur
dioxide ($SO_2$) and 21% for nitrogen oxides ($NO_x \equiv NO+NO_2$) from 2013 to 2017 (Zhang et al.,
2019; Zheng et al., 2018). Correspondingly, annual mean $PM_{2.5}$ concentrations in China
decreased by 30~50% from 2013 to 2017 (Ding et al., 2019; Geng et al., 2021; Li et al.,
2019), avoiding 64 thousand (6.8%) premature deaths. Despite these remarkable achievements,
population-weighted mean $PM_{2.5}$ concentration in China was still as high as 42.1 µg m$^{-3}$ in 2017
(with 2.1 million associated premature deaths; Geng et al., 2021a), far exceeding the newly-
revised threshold of 5 µg m$^{-3}$ in the World Health Organization (WHO) Air Quality Guidelines
(WHO, 2021). In 2020, the Chinese government further launched the "Beautiful China" strategy,
which requires an annual mean $PM_{2.5}$ concentration of $\leq$ 35 µg m$^{-3}$ in all cities by 2035. Yet,
nearly 30% of cities in China exceeded that standard based on the 2021 national observation data
(Ministry of Ecology and Environment, MEE, 2021). Thus, further improvement on air quality
is pressing. However, air quality management has been progressively challenging with the
widespread of end-of-pipe control technologies in dominant sources (industrial and power
sectors; Xing et al., 2020) and the exhausting benefits of such technologies (Geng et al., 2021).
Hence, more comprehensive and in-depth understanding of Chinese $PM_{2.5}$ pollution sources is
urgently needed to help prioritize increasingly limited resources for accurate and effective
mitigation action.

Recent research on the sources of $PM_{2.5}$ pollution in China has focused mostly on China's
domestic anthropogenic emissions (An et al., 2019; Cheng et al., 2021b; Meng et al., 2019; Tang
et al., 2022; Tong et al., 2018; Xing et al., 2020). A number of studies have explored how China
could further reduce its own emissions of air pollutants through a wide range of energy
transformation scenarios to achieve co-benefits of air quality improvement and climate
mitigation (Cheng et al., 2021; Peng et al., 2018; Tong et al., 2018, 2020; Xing et al., 2020).
Studies have also investigated factors hindering the effectiveness of emission reductions on air
quality improvement in China, such as excess ammonia emissions (Bai et al., 2019; Gu et al.,
2021; Yan et al., 2021) and enhanced atmospheric oxidizing capacity associated with $NO_x$
emission reductions in recent years (Huang et al., 2021; Le et al., 2020; Ren et al., 2021; Zang et
al., 2022). A few works have studied the inter-provincial transport of pollution across China and
found that the contribution of inter-provincial transport to $PM_{2.5}$ concentrations in the most
severely polluted regions might exceed that of local emissions (Wang et al., 2022).



However, these previous studies have largely neglected or simplified the transboundary
transport of pollutants from foreign countries to China. This is likely due to the perception that
the relatively short lifetime of $PM_{2.5}$ (a few days) does not permit long-distance transport (Wang
et al., 2019). Only a few have investigated the influence of pollutant emissions from neighboring
countries (Jiang et al., 2013; Koplitz et al., 2017) on China, yet have typically focused on one
particular sector, such as biomass burning emissions from South Asia (Jiang et al., 2013) or coal
emissions from Southeast Asia (Koplitz et al., 2017). A comprehensive assessment of
transboundary $PM_{2.5}$ pollution in China from foreign sources is lacking. In contrast, studies on
the transboundary $PM_{2.5}$ pollution from China to neighboring countries have received
considerable attention (Choi et al., 2019; Jiang et al., 2013; Kurokawa and Ohara, 2020; Park et
al., 2014). This contrast is likely due to another perception that China's domestic emissions far
exceeded those from neighboring countries, such as Korea, Japan, India and the Southeast Asia
(Kurokawa and Ohara, 2020; McDuffie et al., 2020). However, the pollutant emission pattern in
China and neighboring countries may shift in the future. Emissions in China have decreased
considerably (Zheng et al., 2018) and the trend is expected to continue with the launch of
ambitious policies on air pollution (the 2035 "Beautiful China") and climate change (the 2060
carbon neutrality). In contrast, emissions from India and Southeast Asian countries have been
estimated to increase in the future by various projections, given their fast-economic growth and a
lack of clear commitments on either air quality or climate mitigation (IEA, 2021). For example,
Koplitz et al. (2017) revealed that the projected increase of coal emissions in Southeast Asian
countries will lead to 49780 excess deaths per year associated with $PM_{2.5}$ pollution in 2030, with
9000 (18%) of these excess deaths occurring in China. The transboundary pollution from
neighboring countries to China may become increasingly prominent in the future. Thus, an
effective air quality management action for the achievement of the "Beautiful China" target
requires a clear understanding of the current contribution of foreign anthropogenic emissions to
Chinese $PM_{2.5}$ pollution.

A comprehensive assessment of foreign contributions to $PM_{2.5}$ pollution in China relies on a
complex representation of aerosol emissions and chemical reactions across a large spatial
domain. The GEOS-Chem global chemical transport model has been widely applied to $PM_{2.5}$
studies over Asia (i.e., China, India, Southeast Asia, Korea and Japan; Choi et al., 2019; Koplitz
et al., 2017; Miao et al., 2020; Venkataraman et al., 2018; Wang et al., 2004; Zhang et al., 2015),
thereby applicable to such research. Although the model has been extensively validated for total
$PM_{2.5}$ mass concentrations over China using observational data, compositional $PM_{2.5}$ across
China and total $PM_{2.5}$ for other Asian countries are far less evaluated due to scarce observations
(Cheng et al., 2021a; Koplitz et al., 2017; Miao et al., 2020). This limits the credibility of the
model's representation of aerosol emission and chemical reactions across a large domain. Thus, a
more comprehensive evaluation of the GEOS-Chem simulation is needed to support model
estimates of the influence of transboundary pollution on air quality in China.



In this study, we use the GEOS-Chem model to quantify the contributions of foreign anthropogenic emissions to total and compositional $PM_{2.5}$ mass concentrations over China in 2015. We first evaluate our model simulations with comprehensive observations of total and compositional $PM_{2.5}$ concentrations across China and other Asian countries. Then, we quantify the contributions foreign anthropogenic emissions to China $PM_{2.5}$ and compositional concentrations in 2015. Finally, we reveal the physical and chemical pathways leading to such contributions.

## 2. GEOS-Chem simulations

We conducted a series of simulations using the GEOS-Chem chemical transport model (v13.2.1; http://www.geos-chem.org) to 1) represent 2015 $PM_{2.5}$ and composition concentrations over Asia, 2) quantify the contributions of foreign anthropogenic emissions to total and compositional $PM_{2.5}$ concentrations over China, and 3) understand the role and the mechanisms of direct transport and chemical interactions in transboundary pollution in China. Simulation configurations are summarized in Table 1 and are elaborated as the following.

2.1 The GEOS-Chem simulation of ground-level $PM_{2.5}$

We used the flex-grid capability of the GEOS-Chem classic model v13.2.1 to simulate aerosol concentrations over Asia and the adjacent area (11° S–60° N, 30°–150° E; Figure 2a) at a horizontal resolution of 0.5° ×0.625° and at 47 vertical levels between the surface and ∼ 0.01 hPa. The lowest vertical layer has a thickness of about 130 m. We regard the pollutant concentrations in this layer as "ground-level". Detailed descriptions of the flex-grid setup can be found at http://wiki.seas.harvard.edu/geos-chem/index.php/FlexGrid. Our flex-grid domain extended the traditionally-defined nested Asia domain (11° S–55° N, 60°–150° E) in the model (Figure 2a) to better represent the transport of anthropogenic pollutants from Central Asia to China that has not been studied yet. Our simulations were driven by assimilated meteorological data from MERRA-2 provided by the Global Modeling and Assimilation Office (GMAO) at NASA Goddard Space Flight Center. Convective transport in the model was computed from the convective mass fluxes in the meteorological archive as described by Wu et al. (2007). A non-local scheme was used to represent vertical mixing within the planetary boundary layer (PBL), as it accounts for different states of mixing based on the static instability (Lin and McElroy, 2010). Boundary conditions were archived from global simulations at a resolution of 2°× 2.5°. We spun up every simulation for 1 month to remove the effects of initial conditions.

GEOS-Chem simulates $PM_{2.5}$ concentrations as the sum of sulfate ($SO_4^{2-}$), nitrate ($NO_3^-$), ammonium ($NH_4^+$), organic aerosol (OA ≡ primary OA + secondary OA), black carbon (BC), fine dust and fine sea salt component concentrations. The sulfate−nitrate−ammonium (SNA)



aerosol system was simulated following Fountoukis and Nenes (2007) and Park et al. (2004),
including heterogeneous chemistry with dinitrogen pentoxide ($N_2O_5$) uptake by aerosol, and
hydroperoxyl radical ($HO_2$) uptake by aerosol. Gas−aerosol partitioning of SNA was simulated
by the ISORROPIA II thermodynamic equilibrium scheme (Pye et al., 2009). We used a simple
scheme to represent secondary organic aerosol formation (Heald et al., 2012) and used a spatially
resolved ratio to calculate organic mass from organic aerosol concentrations (Philip et al., 2014).
Natural dust simulation followed the Mineral Dust Entrainment and Deposition (DEAD) scheme
(Fairlie et al., 2007). Sea salt aerosol simulation was described in Jaeglé et al. (2011). Dry
deposition of gases and particles followed a standard resistance-in-series scheme, with updates
from Jaeglé et al. (2018). Wet deposition was described in Liu et al. (2001), Wang et al. (2011)
and Wang et al. (2014), with updates from Luo et al., (2020) that included a faster below-cloud
scavenging of $HNO_3$. We calculated the simulated $PM_{2.5}$ and composition concentrations at 35%
relative humidity (RH) for consistency with ground-based measurements.
2.2 Emissions for baseline simulation
We conducted a baseline simulation ("Base" run in Table 1) for 2015 January, April, July,
October and treated the mean of the four months as annual mean. Our simulations were all at a
resolution of 0.5° ×0.625°, unless otherwise specified. The baseline simulation used emissions as
described below.
Anthropogenic emissions for China were taken from the Multi-resolution Emission Inventory
(MEIC) for 2015 (Zheng et al., 2018), and for the rest of the world were taken from the
Community Emissions Data System (CEDS) version 2 for 2015
(https://data.pnnl.gov/dataset/CEDS-4-21-21). Other emissions were default in GEOS-Chem.
Fine anthropogenic fugitive dust emissions from combustion and industrial sources for countries
except China (FR_AFCID) were taken from Philip et al. (2017), and from the MEIC inventory
for China (CH_AFCID). Aircraft emissions were from the Aviation Emissions Inventory Code
(AEIC) inventory (Stettler et al., 2011). Natural emissions include lightning $NO_x$ from Murray et
al. (2012), soil $NO_x$, biogenic non-methane volatile organic carbons (NMVOCs) and sea salt
from off-line emissions developed by Weng et al. (2020), biomass burning emissions from the
Global Fire Emissions Database version 4 (GFED4; Randerson et al., 2015), volcano emissions
from Fisher et al. (2011), marine dimethyl sulfide (DMS) emissions from Breider et al. (2017)
and dust emissions using the Mineral Dust Entrainment and Deposition (DEAD) scheme (Zender
et al., 2003).
2.3 Sensitivity simulations for the contributions of foreign anthropogenic emissions to China
$PM_{2.5}$ and composition concentrations



We quantified contributions of foreign anthropogenic emissions to China total and
compositional PM$_{2.5}$ concentrations by taking the difference of the baseline simulation ("Base"
run in Table 1) and a sensitivity simulation that excluded foreign anthropogenic emissions from
the baseline simulation ("CHAnth" run in Table 1). Such a foreign contribution is referred to as
"FR_total".
We further conducted sensitivity simulations to attribute the transboundary pollution to 1)
direct transport of foreign PM$_{2.5}$ to China and 2) chemical interactions between transported
foreign pollutants and Chinese emissions. We quantified the contribution of direct transport in
transboundary pollution (referred to as "FR_transport") by taking the difference of a sensitivity
simulation that excluded China anthropogenic emissions ("FRAnth" run in Table 1) and another
sensitivity simulation that excluded both China and foreign anthropogenic emissions ("NoAnth"
run in Table 1). Transboundary pollution through chemical interactions with China's local
emissions (referred to as "FR_chemistry") were calculated as the differences between total
foreign anthropogenic contributions (FR_total) and direct transport contributions (FR_transport).
We conducted sensitivity simulations to understand main pollutants driving the chemical
interactions of transboundary pollution with Chinese emissions. Specifically, we quantified the
contributions of foreign anthropogenic emissions of NMVOCs to O$_3$, NO$_3$ and N$_2$O$_5$, HNO$_3$ and
NO$_3^-$ concentrations in China. Chemical interactions between foreign anthropogenic emissions of
NMVOCs and China domestic emissions of pollutants were calculated as the differences of total
contributions by foreign anthropogenic emissions of NMVOCs ("Base" –
"No_FRAnthNMVOCs" runs in Table 1) and the direct transport share of the total contributions
("FRAnthNMVOCs" – "NoAnth" runs in Table 1).
To reduce computational costs, we conducted NMVOCs-related sensitivity simulations at a
resolution of 2°× 2.5° (Table 1). The differences of PM$_{2.5}$ over eastern China between 2°× 2.5°
and 0.5° ×0.625° resolutions is about 5% (3 μg m$^{-3}$). Compositional differences are within 20%,
with the largest difference in nitrate (2.6 μg m$^{-3}$; 18%) and the lowest in black carbon (0.1 μg m$^{-3}$
$^{3}$; 4%). These differences are within the reasonable range associated with spatial resolutions.
**3. Ground-level observations of PM$_{2.5}$ and compositions in China and other Asian**
**countries**
We evaluated the modeled PM$_{2.5}$ and composition concentrations in the base year (2015) using
ground-level PM$_{2.5}$ observations from the network of China National Environmental Monitoring
Center (CNEMC), the literature search and the World Health Organization (WHO) database.
Ground-level PM$_{2.5}$ observations were obtained from the CNEMC network
(http://106.37.208.228:8082/). We used hourly measurements for 2015 January, April, July and



October. $PM_{2.5}$ mass concentrations were measured using the micro-oscillating balance method
or the β- absorption method (Zhang and Cao, 2015). We further applied quality controls to
hourly CNEMC data. Specifically, we removed a day if there were < 14 valid data within the day
and removed a month if there were < 25 days of valid data within the month. These in whole
removed 12% $PM_{2.5}$ hourly data, and finally retained observations for 1179 sites in 314 cities for
model evaluation (Fig. S2). To compare with the GEOS-Chem simulated $PM_{2.5}$ concentrations,
we calculated the grid-averaged and monthly-averaged $PM_{2.5}$ concentrations from the CNEMC
to match spatially and temporally with the model.
We collected compositional $PM_{2.5}$ observations from publicly available studies, as shown in
Table S1. We selected observations that spun at least one-year or seasonal/monthly
measurements centered at January, April, July or October to match our model simulations. A
total of 56 observation data from 17 cities in 16 provinces for 2014–2016 were collected for
major $PM_{2.5}$ chemical composition, including sulfate, nitrate, ammonium, organic aerosol (OA),
and black carbon. We sampled the GEOS-Chem simulated concentrations from locations and
periods (monthly or annual) of observations for evaluation.
To evaluate the modelled $PM_{2.5}$ concentrations outside China, we collected $PM_{2.5}$
measurement data for 2013–2016 from the WHO ambient air pollution in cities database
(https://www.who.int/data/gho/data/themes/air-pollution). This database provides annual average
$PM_{2.5}$ concentrations for more than 500 cities globally. We retained $PM_{2.5}$ measurements that
were directly measured and excluded data inferred from $PM_{10}$ concentrations. We also retained
the sites within our simulation domain and the grid cells where anthropogenic emissions
exceeded natural emissions (by comparing $PM_{2.5}$ concentrations in "FRAnth" and "NoAnth"
runs in Table 1). A total of 83 sites covering 10 Asian countries (Bangladesh, Indonesia, India,
Japan, Korea, Malaysia, Myanmar, Philippine, Thailand and Vietnam) were finally selected in
our study as shown in Fig. 2a. Because of the differences in measurement approaches between
jurisdictions, and the absence of details regarding measurement data (Brauer et al., 2016), we
compared annual average $PM_{2.5}$ concentrations in the WHO dataset with those estimated from a
hybrid of satellite observations, a chemical transport model and ground-based measurements
(van Donkelaar et al., 2021). We removed the sites if the differences were more than 40%. The
final sites retained in our study are shown in Fig. 2b.
**4. $PM_{2.5}$ and compositions concentrations across China and other Asian countries**
Figure 1 shows modelled and observed annual mean concentrations of $PM_{2.5}$ and compositions
across China. The modelled total and compositional $PM_{2.5}$ are for 2015. Observations of total
$PM_{2.5}$ are for 2015 and of compositions are for 2014–2016. The spatial distribution of modelled
$PM_{2.5}$ exhibits a broad high across the eastern China, driven primarily by sulfate ($SO_4^{2-}$), nitrate
($NO_3^-$), ammonium ($NH_4^+$), and organics (OA). $PM_{2.5}$ over the Sichuan Basin is also high,



contributed mostly by sulfate, ammonium and organics. The enhanced $PM_{2.5}$ concentrations over
the west is dominated by mineral dust. Compared to observations, the modelled $PM_{2.5}$ well
reproduces the spatial variation of $PM_{2.5}$ across China, with a correlation coefficient (r) of 0.73
and a normalized mean bias (NMB) of 15.7%. The slight overestimation is primarily contributed
by sites in the Sichuan Basin (Figure S2a), where local emissions (anthropogenic and vegetation
emissions; Wang et al., 2018), meteorological conditions (humid and stagnant; Chen et al., 2014;
Liao et al., 2017), combined with the special terrain (plain surrounded by hills; Chen et al., 2014;
Wang et al., 2017) make it hard for the model to represent aerosol processes, especially the
secondary aerosol formation process (Liao et al., 2017; Tao et al., 2017; Wang et al., 2018).
We further evaluate our modelled compositional $PM_{2.5}$ with observations from 56 sites across
16 provinces and municipalities in China to better understand the performance of the simulation.
The scatterplots of composition comparison in Fig. 1 show a mixture of annual and monthly data
depending on the availability of observations data from the literature. We find that the modelled
compositional $PM_{2.5}$ reasonably reproduce the vast spatial variation of $PM_{2.5}$ composition from
observations. Sulfate and nitrate simulations are particularly improved over previous model
studies where sulfate was significantly underestimated (NMB~-40%) and nitrate was
substantially overestimated (NMB~80%; Gao et al., 2018; Miao et al., 2020). The better
representation of nitrate is owing to the faster nitrate removal in the Luo et al. (2020) deposition
scheme. Organics are also well reproduced by the model (NMB=4.7%), suggesting the
effectiveness of the simple SOA scheme in representing total SOA mass. Ammonium and black
carbon show relatively large discrepancies (NMB = 24.9% and -12.8%, respectively), potentially
reflecting model biases in chemical reactions or gas-particle partition for ammonium formation
(Miao et al., 2020) and emission inventories for BC (Zhang et al., 2019). In addition, outstanding
differences in observations approaches (i.e., thermal, optical or incandescence measurements for
black carbon; different relative humidity in measurements; Bond et al., 2013; Snider et al., 2016)
across literature is another major reason for the discrepancies.
The evaluation of modelled $PM_{2.5}$ concentrations for other Asian regions has been rarely
conducted due to limited observations (Koplitz et al., 2017). Here, we compare our modelled
$PM_{2.5}$ concentrations with observations from 10 Asian countries around China to understand the
model performance in regions whose pollution could influence China through transboundary
transport. Figure 2 shows good agreement between modelled and observed total $PM_{2.5}$ mass
across countries, with a correlation coefficient of 0.76 and a NMB of 3.5%, despite large
uncertainties in the measurements collected by different countries. There is an underestimate at
coastal sites (e.g., in Philippine; Fig. 2a) where sea salt aerosols potentially make larger
contributions than our simulations. Specifically, our simulated $PM_{2.5}$ is very consistent with
observations in the Southeast Asia (NMB = 2.7%; including Indonesia, Myanmar, Philippines,
Thailand and Vietnam), India (3.8%) and other countries in South Asia (NMB=7.9%; including
Bangladesh, Bhutan, Maldives, Nepal, Pakistan, Sri Lanka). Simulations over Japan and South

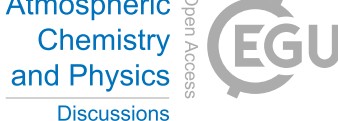

Korea show relatively larger bias (NMB = 17.4% for Japan and 40% for South Korea). Zhai et
al. (2021) attributed the 43% bias in the GEOS-Chem simulation of surface $PM_{2.5}$ over South
Korea in their study to nighttime nitrate formation, although their updates of faster below-cloud
scavenging of $HNO_3$ (Luo et al., 2020) corrected the overall nitrate bias in East Asia.
**5. Contributions of foreign anthropogenic emissions to total and compositional $PM_{2.5}$ in**
**China**
Figure 3 shows the contributions of foreign anthropogenic emissions to total and
compositional $PM_{2.5}$ concentrations over China in 2015. On the national level, foreign
anthropogenic emissions contribute about 2.4 $\mu g\ m^{-3}$ $PM_{2.5}$ to China in 2015, accounting for
6.2% of the national average $PM_{2.5}$ concentration. The foreign influence exhibits prominent
spatial heterogeneity, with the largest contribution of 5.0 $\mu g\ m^{-3}$ $PM_{2.5}$ (8%) to the eastern China
(outlined in Fig. 3; including Anhui, Hebei, Henan, Jiangsu, Liaoning, Shandong, Beijing and
Tianjin). Considering the WHO newly-revised guideline for $PM_{2.5}$ annual exposure level ($\leq 5\ \mu g$
$m^{-3}$), foreign anthropogenic emissions alone would make $PM_{2.5}$ concentrations over the eastern
China reach the WHO standard, threating the health of nearly 500 million residents there.
Transboundary pollution is also outstanding along the southwestern border, contributing 4.9 $\mu g$
$m^{-3}$ $PM_{2.5}$ (18%) to Yunnan province, mostly driven by anthropogenic emissions from South Asia
(i.e., India).
The transboundary pollution over the eastern and the southwestern China is contributed by
different chemical components of $PM_{2.5}$, as shown in Fig. 3. The eastern China is mainly driven
by nitrate and ammonium, explaining 70% of transboundary $PM_{2.5}$. Particularly, 18% of nitrate
and 12% of ammonium concentrations over the eastern China in 2015 are driven by
transboundary pollution. Leibensperger et al. (2011) and Koplitz et al. (2017) found similar
nitrate enhancement yet to a much lesser extent (< 0.2 $\mu g\ m^{-3}$). They attributed the nitrate
enhancement to the increase in ozone that speeded up the rate at which Chinese local $NO_x$
emissions were converted to nitrate. We will discuss the mechanism in more detail in the next
section. The sulfate contribution is very small (< 0.5 $\mu g\ m^{-3}$). Leibensperger et al. (2011)
proposed a reason that $H_2O_2$ (the key oxidant for sulfate formation) is abundant most of the year
over the eastern China, thereby insensitive to additional transboundary source. When $H_2O_2$-
limited conditions prevail in winter, cloud cover is infrequent, limiting the in-cloud oxidation of
$SO_2$. Thus, the influence of transboundary pollution to sulfate over the eastern China is weak.
The transboundary $PM_{2.5}$ over the southwestern China is primarily contributed by organics (1.8
$\mu g\ m^{-3}$; 37% of transboundary $PM_{2.5}$), which is consistent with previous studies that found
massive biomass burning emissions in South Asia contributed considerable organics to the
southern China (Jiang et al., 2013). Sulfate contributes 27% of transboundary $PM_{2.5}$ over the
southwestern China, where the inflow of hot and humid atmosphere from the South Asia
facilitates the in-cloud formation of sulfate (Jiang et al., 2013). In addition, anthropogenic





fugitive dust emissions from foreign countries make an influence to China PM$_{2.5}$, accounting for
14% of the PM$_{2.5}$ increase in both the eastern and southwestern regions, as shown in Figure S3.
4       We further investigate the seasonal variation of transboundary pollution in China to
understand potential sources of foreign contributions. Figure 4 presents the seasonal
enhancement of PM$_{2.5}$, nitrate and ammonium over China in 2015 driven by foreign
anthropogenic emissions. Transboundary pollution of PM$_{2.5}$ over the eastern China is the largest
in January (6.9 µg m$^{-3}$) and gradually decreases to the smallest in July (2.7 µg m$^{-3}$). Affected
regions also change prominently with seasons. In January, the boundary of transboundary PM$_{2.5}$
larger than 9 µg m$^{-3}$ extends to the south of the eastern China domain outlined in Fig. 4, whereas
in July, that boundary shrinks to a much smaller region along the east coast.  These seasonal
characteristics of transboundary PM$_{2.5}$ are similar to those of nitrate and ammonium. In January,
68% transboundary PM$_{2.5}$ over the eastern China is contributed by nitrate (4.7 µg m$^{-3}$) and 19%
by ammonium (1.3 µg m$^{-3}$), yet these fractions decrease to 11% for both nitrate and ammonium
in July. The transboundary nitrate for the majority of the eastern China exceeds 5 µg m$^{-3}$ in
January, yet decreases to less than 1 µg m$^{-3}$ and even negative in July. These prominent seasonal
variations of transboundary PM$_{2.5}$, nitrate and ammonium reflect different processes controlling
the transboundary pollution in winter and summer in China.
**6. Physical and chemical mechanisms of foreign anthropogenic contributions to PM$_{2.5}$**
**over eastern China**
We investigate the relative importance of direct transport and chemical interactions in driving
the considerable transboundary pollution over the eastern China in January and July, as shown in
Figure 5. In January, the transboundary PM$_{2.5}$, nitrate and ammonium are predominantly (71–
97%) driven by chemical interactions, suggesting that the transboundary pollution in winter over
the eastern China is not through direct transport of nitrate and ammonium, but through chemical
interactions between directly transported precursors from foreign countries and local emissions
in China. In July, however, nearly all transboundary PM$_{2.5}$ over the eastern China is driven by
direct transport, with 30% of the direct transport contributed by anthropogenic fugitive dust from
foreign countries (Fig. S3). The transboundary nitrate is still primarily driven by chemical
interactions in July (89%), yet the magnitude is too small to substantially affect total PM$_{2.5}$.
We further explore the changes in key chemical species for nitrate formation to understand the
chemical mechanism driving the considerable nitrate enhancement over the eastern China. Figure
6 shows the contributions of transboundary pollution to concentrations of precursor gases (NO$_x$),
oxidants (O$_3$, N$_2$O$_5$ and NO$_3$) and oxidized products (total inorganic nitrate including gas-phase
HNO$_3$ and particulate NO$_3^-$) in nitrate chemistry over the eastern China in January and July. Fig.
6 (top) shows that, in January, 3.6 ppb O$_3$ are directly transported from foreign countries, yet
52% (1.9 ppb) of which further undergo chemical reactions, leading to a drop of NO$_x$ (-1.8 µg N



m$^{-3}$) and an increase of $N_2O_5+NO_3$ and $HNO_3+NO_3^-$ (0.88 µg N m$^{-3}$) concentrations over the
eastern China. The presence of excess ammonia over the eastern China due to the reduction of
$SO_2$ (Liu et al., 2018, 2019) further partitions nearly all $HNO_3$ gas to into particulate nitrate
(nitrate ammonium, $NH_4NO_3$), leading to about 1 µg N m$^{-3}$ (or 2.6 µg m$^{-3}$) nitrate increase there.
These results reveal that, in January, the additional $O_3$ from transboundary sources interact with
local emissions of $NO_x$ over the eastern China and promote the nitrate formation, which
otherwise would be limited by the lack of $O_3$ (Jin and Holloway, 2015; Li et al., 2018; Wang et
al., 2017). Fig. 6 (top) further reveals that 93% of the direct transboundary transport of ozone and
eventually 72% of the nitrate increase over the eastern China is contributed by foreign
anthropogenic emissions of NMVOCs. Thus, the transboundary transport of ozone precursors
(primarily NMVOCs) combined with high domestic emissions of $NO_x$ and ammonia in winter
makes the considerable increase of nitrate concentrations over the eastern China.
In July, however, although foreign sources contribute about 2 ppb $O_3$ to the eastern China
through direct transport, they hardly lead to much difference in nitrate concentrations (Fig. 6
bottom). This is because that a lack of excess aerosols (compared to winter) limits the
transformation of $N_2O_5$ and $NO_3$ to $HNO_3$ on aerosol surface. In addition, the oxidation of $NO_x$
by OH is sufficiently fast in summer and the abundance of OH over the eastern China in summer
makes the $NO_x$ oxidation process insensitive to the added $O_3$ from foreign countries. Therefore, a
lack of excess aerosols and the abundance of OH in summer makes the transboundary transport
of ozone precursors minor in contributions to nitrate concentrations over the eastern China.
**7. Conclusions**
An effective air quality improvement action requires an accurate understanding of $PM_{2.5}$
sources. This work complements our understanding of $PM_{2.5}$ sources in China by investigating
the influence of foreign transboundary transport through the GEOS-Chem simulation. Our
extensive and comprehensive evaluation of the GEOS-Chem model for $PM_{2.5}$ total and
compositional mass concentrations in China and 10 additional Asian countries showed a
reasonable consistency with observations. Based on model simulations, we found that foreign
anthropogenic emissions played an important role in Chinese $PM_{2.5}$ pollution, because of direct
aerosol transport and, more importantly, chemical interactions between transboundary pollutants
and China's local emissions. Over the eastern China, the transport of NMVOCs from foreign
anthropogenic emissions increased the background $O_3$ level by 3.3 ppb, which combined with
high local emissions of $NO_x$ and ammonia led to a nitrate enhancement of 2.6 µg m$^{-3}$ in January.
Over the southwestern China, transboundary transport contributed 18% $PM_{2.5}$ to Yunnan
province in 2015, mostly driven by the direct transport of aerosols from anthropogenic emissions
in South Asia. There are a few sources of uncertainty in this study, for example the wet
deposition of nitrate and the simplified secondary organic aerosol formation scheme, but they do
not manifest themselves as systematic biases.



In light of the physical and chemical mechanisms of transboundary pollution in China, further improvements of air quality for the "Beautiful China" target requires different emission reduction strategies for different regions. Over the eastern China, reductions of both foreign and domestic anthropogenic emissions can reduce transboundary $PM_{2.5}$ pollution to China, since a considerable amount of transboundary transported $PM_{2.5}$ is formed through the chemical interactions between pollutants from both sources. Over the southwestern China, foreign emission reductions will be necessary for improving air quality there since the transboundary pollution was primarily through the direct transport of aerosols from foreign countries. Given the declining trend of Chinese anthropogenic emissions in the present and the future, the likely rising emissions from adjacent countries in the future could become an increasingly important problem for Chinese air quality protection if no action is taken to avoid emissions increases in adjacent countries. Our study points to the need to carefully consider the potential influence of transboundary pollution when making the long-term air quality improvement strategies in China. Future studies could further investigate the impact of transboundary pollution transport to China under future emission scenarios, or extend this study to other regions in the world where transboundary pollution could potentially increase domestic $PM_{2.5}$ pollution through nitrate chemistry.

**Data availability**

Data presented in this paper are available upon request to the corresponding author.

**Author contributions**

J.L. led the study. J.X. and J.L. designed the study. J.X. performed the model simulations and conducted the data analysis. J.A. collected observation data of $PM_{2.5}$ compositions from the literature. H.K. processed $PM_{2.5}$ observations from the CNEMC website. J.X. wrote the manuscript with inputs from J.L. All authors commented on the manuscript.

**Competing interests**

The authors declare that they have no conflict of interest.

**Acknowledgements**

The author would like thank CNEMC staff for providing $PM_{2.5}$ measurements across China and WHO staff for providing global $PM_{2.5}$ measurements. This work was supported by the National Natural Science Foundation of China (42075175), the China Postdoctoral Science Foundation (2021M700191) and the Peking University Boya Postdoctoral Fellowship.





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

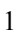

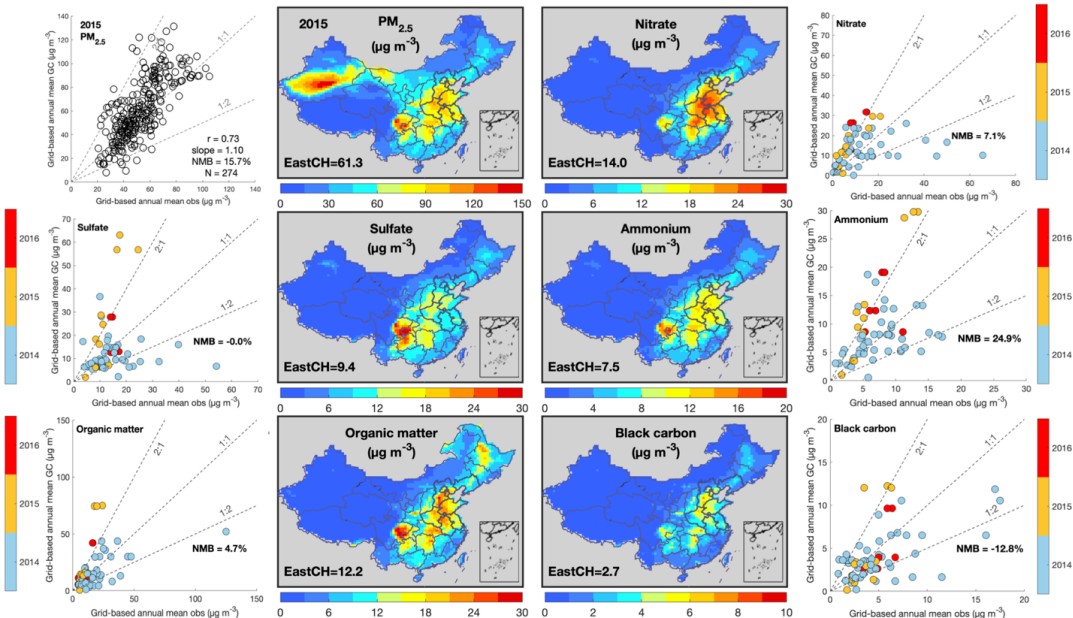

**Figure 1**. Total and compositional PM$_{2.5}$ concentrations in China. Spatial distributions of PM$_{2.5}$
and composition concentrations are annual mean concentrations simulated by the GEOS-Chem
model for 2015 January, April, July and October with a resolution of 0.5° x 0.625°. We regard
the mean of the four months as annual mean. Thick black lines outline the eastern China
discussed in this work. Text in the bottom left corner of each map refers to mean concentrations
($\mu$g m$^{-3}$) over the eastern China. The scatterplot of PM$_{2.5}$ compares the simulated annual mean
concentrations with collocated and coincident observations for 2015 from the CNEMC network.
Scatterplots of composition compares both annual and monthly concentrations of the observed
and the coincident simulated concentrations according to the availability of observations from
the literature for 2014-2016.

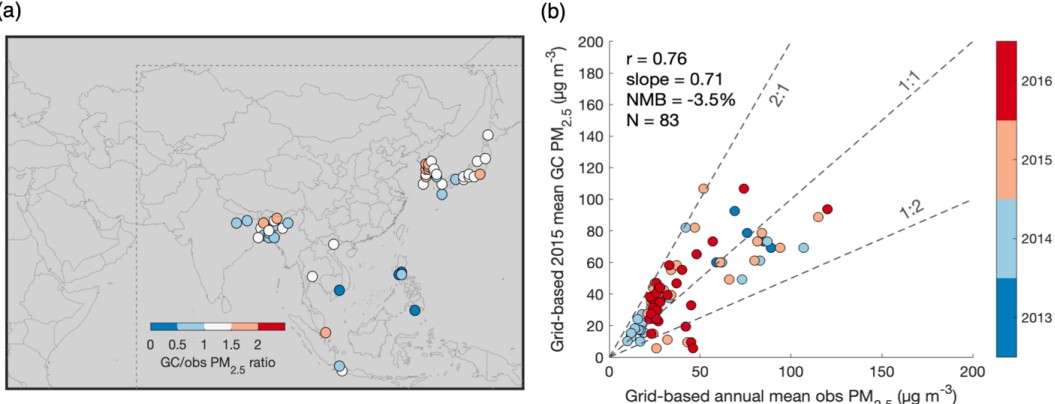

**Figure 2**. Annual mean PM$_{2.5}$ concentrations at anthropogenic emission dominated sites in the
simulation domain outside China. (a) The spatial distribution of simulated and observed PM$_{2.5}$
concentration ratios. The simulated concentration at each measurement site represents the 0.5°x
0.625° grid cell covering that site. Dashed lines represent the default nested Asia domain (11° S–
55° N, 60°–150° E) in the model. The flex-grid domain in our study is shown as the entire
domain of the map. (b) Scatterplot comparing simulated concentrations for 2015 with collocated
observations from the WHO for 2013–2016.

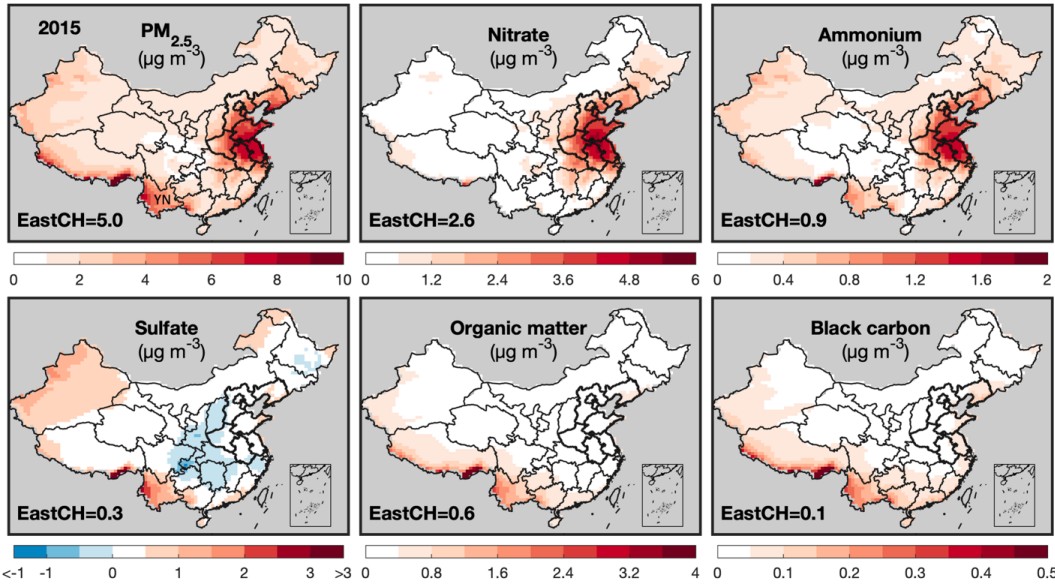

**Figure 3**. Simulated annual mean contributions of foreign anthropogenic emissions to China's
total and compositional PM$_{2.5}$ concentrations in 2015. Thick black lines outline the eastern China
discussed in this work. Text in the bottom left corner of each panel refers to mean concentrations
(µg m$^{-3}$) over the eastern China contributed by foreign anthropogenic emissions. The foreign
impact on China's anthropogenic dust concentrations are shown in Figure S3. YN in the $PM_{2.5}$
subplot refers to the location of Yunnan province.
**Figure 4**. Same as Fig. 3, but for January, April, July and October as denoted by text in the top
left corner of each row.

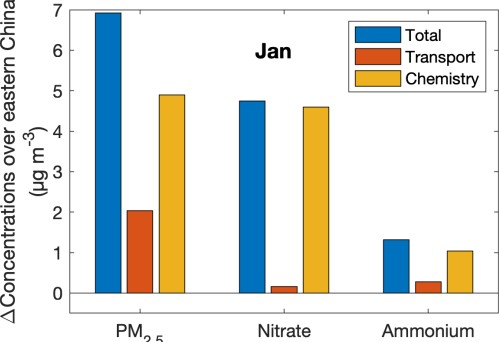
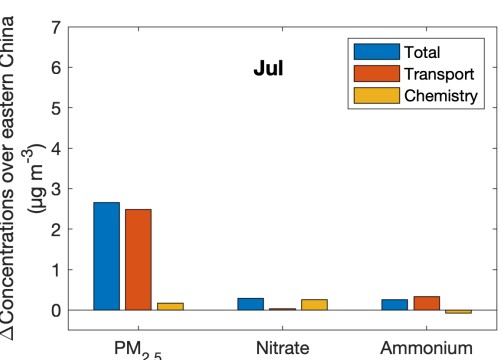

**Figure 5**. Foreign anthropogenic contributions to PM$_{2.5}$, nitrate and ammonium concentrations in January and July over the eastern China. Total concentration contributions of foreign anthropogenic emissions are split into contributions from direct transport and chemical interactions according to the legend. The contributions to other PM$_{2.5}$ components are shown in Figure S4.

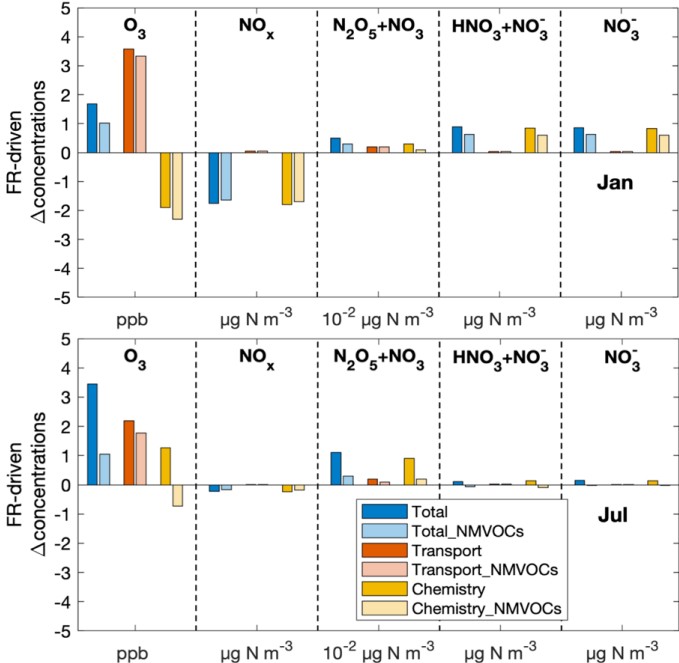

**Figure 6**. Contributions of foreign anthropogenic emissions of total aerosols and NMVOCs to atmospheric oxidizing capacity and oxidation products over the eastern China as simulated by the GEOS-Chem model for 2015 January (top) and July (bottom). Total foreign anthropogenic contributions are split into contributions from direct transport and chemical interactions according to the legend. O$_3$ concentrations are 24-hour average concentrations. Nitrogen-related species are presented in the unit of µg N m$^{-3}$ for the convenience of nitrogen budget calculation.



1 **Table 1**. Configuration summary of GEOS-Chem simulations in this study. Emission
2 abbreviations are elaborated in Section 2.2.

| Foreign contribution type | | | FR_Total | | FR_Transport | | FR_NMVOCs | |
|---|---|---|---|---|---|---|---|---|
| **Simulation name** | | | Base | CHAnth | FRAnth | NoAnth | No_FRAnthNMVOCs | FRAnthNMVOCs |
| **Emissions** | CHAnth (MEIC+CH_AFCID) | | Y | Y | N | N | Y | N |
| | FRAnth (CEDS+FR_AFCID) | VOCs | Y | N | Y | N | N | Y |
| | | OTR[1] | | | | | Y | Y |
| | Other (Shipping+aircraft+natural) | | Y | Y | Y | Y | Y | Y |
| **Resolution** | | | 0.5° x 0.625° | | 0.5° x 0.625° | | 2° x 2.5° | |
| **Simulation period (year/month)** | | | 2015/1,4,7,10 | | 2015/1,4,7,10 | | 2015/1,7 | |
| **Met fields** | | | MERRA2 | | MERRA2 | | MERRA2 | |

3 [1]OTR refers to other species including $SO_2$, $NO_x$, $NH_3$, BC, OC, CO, etc.