# Peer review of "Foreign emissions exacerbate PM2.5 pollution in China through nitrate chemistry 1 2 3 Jun-Wei Xu1, Jintai Lin1\*, Gan Luo2, Jamiu Adeniran1, Hao Kong1 4 5 1Laboratory for Climate and Ocean–Atmosphere Studies, Department of"

_Atmospheric Chemistry and Physics, 2022_

## Author Response (AR1)

**Authors' Response to Comments from Referee #1**

This is an interesting, well-executed analysis with clear policy relevance. In my view this paper could be published as is, once the minor comment on Figure 1 is corrected (see below).

**Response: we sincerely thank the Referee #1 for taking the time to review our paper and for providing constructive suggestions for improvement. Reponses to these comments are provided below.**

General Questions

Did you also assess how the comparisons with observations shown in Figure 1 vary by season? My understanding is that the regional transport patterns can vary substantially throughout the year, e.g. with the north-south migration of the ITCZ. Do you have enough ambient data to characterize the ability of GEOS-Chem to capture some of these seasonal patterns more specifically?

**Response: we added the seasonal evaluation of simulated PM$_{2.5}$ with observations as shown in Fig. S2. In the main text P8, we added the following description: "The seasonal evaluation of simulated PM$_{2.5}$ are shown in Fig. S2 and exhibits a good consistency with observations across seasons (r = 0.61~0.77), demonstrating the model's capability in capturing the seasonal pattern of aerosol processes."**

Line-by-line

Figure 1. Hard to see outline of East China (easier to see in Figure 3)

**Response: revised the outline of eastern China in Fig. 1.**

**Authors' Response to Comments from Referee #2**

This manuscript quantifies the contributions of foreign anthropogenic emissions to total and compositional PM2.5 mass concentrations over China. The topic is of interest to the community. It is also well written. But several problems should be solved before publication.

**Response: we thank the Referee #2 for providing constructive suggestions for our study. Reponses to these comments are provided below.**

1. Since you conclude that foreign anthropogenic emissions play an important role in Chinese PM5 pollutions, I wonder which source makes the largest contributions. You'd explore the contribution of different sources for reference of emission reductions.

   **Response: Thanks for this insightful suggestion. We conducted additional simulations to quantify sectoral contributions of foreign anthropogenic emissions to China PM$_{2.5}$ concentrations. The result was included as Figure S4. We added the following descriptions to the Method, the Results and the Conclusion parts to reflect this revision:**

   **P6: "We also conducted simulations to quantify the sectoral contributions of foreign anthropogenic emissions to China's PM$_{2.5}$ concentrations. Sectoral contributions were calculated by taking the difference of a simulation that included one sector of foreign anthropogenic emissions (agriculture, industry, energy, traffic, residential combustion, solvent use, waste burning) at a time one and a simulation without foreign anthropogenic emissions ("CHAnth" in Table 1)."**

   **P11: "Source attribution (Fig. S4) reveals that all of the seven major anthropogenic emission source sectors in foreign countries contributed similarly (10-17% for each sector) to China PM$_{2.5}$ in January, yet their relative importance exhibits spatial heterogeneity. Over eastern China, industry and solvent use in foreign countries are the largest sources (likely because of their considerable amount of NMVOCs emissions that increase the atmospheric oxidizing capacity over eastern China, as will be discussed in Section 6), whereas over Yunnan province, residential combustion in foreign countries makes the largest contribution."**

   **P13: "To reduce the transboundary influence driven by chemical interactions, source sectors with considerable emissions of NMVOCs in foreign countries, such as industry and solvent use, are of particular importance."**

2. Maybe the contributions of domestic emissions to neighboring countries should also be explored, then the extent to which the transboundary PM2.5 pollutions from countries contribute to each other has more practical significance to develop a common policy to reduce emissions.

**Response: Thanks for the suggestion. The contributions of China's domestic emissions to neighboring countries have been explored in previous studies (Choi et al., 2019; Jiang et al., 2013; Kurokawa and Ohara, 2020; Park et al., 2014). Our study is one of the few that consider the reverse contribution, which is the contribution of foreign emissions to China. That's exactly where our novelty lies in. We have described that on P3: "In contrast, studies on the transboundary PM$_{2.5}$ pollution from China to neighboring countries have received considerable attention (Choi et al., 2019; Jiang et al., 2013; Kurokawa and Ohara, 2020; Park et al., 2014). The contrast is likely due to another perception that transboundary pollution from foreign countries to China is minor since China's domestic emissions far exceeded those from neighboring countries, such as Korea, Japan, India and the Southeast Asia (Kurokawa and Ohara, 2020; McDuffie et al., 2020). However, the pollutant emission pattern in China and neighboring countries may shift in the future …"**

**In addition, we further investigated the mechanism of the transboundary influence to China, and revealed that chemical interactions between China's domestic emissions and transboundary-transported pollutants (through nitrate chemistry) played a significant role in PM$_{2.5}$ pollution over eastern China. This mechanism has been hardly discussed in previous studies and is therefore the focus of our study.**

**In summary, the reviewer's suggested net contributions would be a good extension to our study, yet beyond the scope of this current work. Thus, we included in the Conclusion section on P13 that "Investigation into the bi-directional PM$_{2.5}$ contributions between countries would also be helpful for developing practical policies for regional cooperation on emission reductions."**

P2: "A few works have studied the inter-provincial transport of pollution across China and found that the contribution of inter-provincial transport to PM2.5 concentrations in the most severely polluted regions might exceed that of local emissions.". Regarding the regional transport, more references and discussions are needed such as Li, et al, 2015. Reinstate regional transport of PM2.5 as a major cause of severe haze in Beijing, Proc Natl Acad Sci USA (PNAS), doi:10.1073/pnas.1502596112. 112(21), E2739–E2740.

**Response: Thanks for providing this reference. We have added the suggested reference and revised the relevant discussion on P2-3 to "A few works have studied the inter-provincial transport of pollution across China and found that the contribution of inter-provincial transport to PM$_{2.5}$ concentrations in the most severely polluted regions (such as Beijing) might have exceeded that of local emissions (Li et al., 2015). In addition, the range of inter-provincial transport of pollution was not confined within city clusters, such as Beijing-Tianjin-Hebei, Yangtze River Delta, and Pearl River Delta, but also extended over a long distance across city clusters (Wang et al., 2022). Yan et al. (2021b) further found that transboundary transport from Asian regions (18.5–19.2%, including Chinese regions outside the Wuhan City Cluster) contributed much more to ozone concentrations in the Wuhan City Cluster in Central China than the transport within the city cluster (2.5–3.1%), highlighting the importance of transboundary transport of pollutants to China."**

3.  P11, "…the transboundary transport of ozone precursors (primarily NMVOCs) combined with high domestic emissions of NOx and ammonia in winter makes the considerable increase of nitrate concentrations over the eastern China.". I believe that this also increased SOA concentrations. Please have some discussions.

**Response: Thanks for pointing this out. As shown in Fig. 3, the contribution of foreign anthropogenic emissions to organic matter in China is minor. This is partly because that the simplified SOA formation scheme in the model may not be able to fully represent the chemical formation of SOA, which is a common issue in chemical transport models (Pennington et al., 2021; Shrivastava et al., 2017). Thus, our discussion on the chemical interactions initiated by the transport of foreign pollution is not focused on SOA.**

**We added the following to P10 to clarify this point: "However, such enhancement over eastern China is not observed for organic matter, partly due to the simplified SOA formation scheme that is not able to fully represent the chemical formation of SOA, which is a common issue in chemical transport models (Pennington et al., 2021; Shrivastava et al., 2017)."**

4.  Several English problems in the text part such as

(1) P4, L6: "of" should be add between "the contributions" and "foreign".

**Response: Done**

(2) P8, L25: "observations approaches" => "observation approaches"

**Response: Done**

(3) P8, L27: "is" => "are"

**Response: Done**

(4) In Table1, for the FRAnthNMVOCs case, I think CHAnth (MEIC+CH_AFCID) should be "Y".

**Response: We turned off China's domestic emissions in the FRAnthNMVOCs case to avoid chemical interactions between China's domestic emissions of pollutants and foreign-emitted NMVOCs, so that the difference between the FRAnthNMVOCs run and the NoAnth run (Table 1) can represent the contributions of foreign NMVOCs to China by direct transport (not by chemical interactions). We revised the description on P6 to make it clear:**

**"We conducted sensitivity simulations to understand main pollutants driving the chemical interactions of transboundary pollution with Chinese emissions. Specifically, we quantified the contributions of foreign anthropogenic emissions of NMVOCs and $NO_x$ to $O_3$, $NO_3$ and $N_2O_5$, $HNO_3$ and $NO3^-$ concentrations in China. Such contributions include both the direct transport of foreign pollutants and chemical interactions between foreign-transported and China domestic emissions of pollutants. We quantified the total contributions by foreign anthropogenic emissions of NMVOCs as the difference between a simulation with full emissions and a simulation that excluded foreign anthropogenic emissions of NMVOCs ("Base" – "No_FRAnthNMVOCs" runs in Table 1). To quantify the direct transport share of the total contributions, we excluded China's domestic emissions to avoid interactions with foreign-transported pollutants, and calculated it as the difference between simulations that included and excluded foreign anthropogenic emissions ("FRAnthNMVOCs" – "NoAnth" runs in Table 1). The contributions of chemical interactions between foreign NMVOCs and China's domestic emissions were quantified as the difference between the total contributions and the direct transport share of the total contributions."**

**Instead of the above-mentioned misunderstanding, we corrected a real mistake in our original FR_NMVOCs runs, which did not much affect the results. In the original simulations, we accidentally included foreign anthropogenic emissions of $NO_x$ in the FR_NMVOCs runs that meant to represent the contributions of foreign anthropogenic emissions of pure NMVOCs to China $PM_{2.5}$ concentrations. We corrected this mistake by excluding foreign anthropogenic $NO_x$ emissions in the FR_NMVOCs runs, and we isolated the influence of foreign anthropogenic emissions of $NO_x$ to China $PM_{2.5}$**

concentrations as additional sensitivity simulations ("FR_NO$_x$" in Table 1 and results are shown in Fig. 6). We added the settings of these new foreign NO$_x$ sensitivity runs to the Method section on P7: "Similarly, the total contributions by foreign anthropogenic emissions of NO$_x$ were calculated as "Base" – "No_FRAnthNO$_x$" runs in Table 1, with the corresponding direct transport share of the total contributions calculated as "FRAnthNO$_x$" – "NoAnth" runs in Table 1, and the chemistry share calculated as the difference between the total and the direct transport share." The result that foreign anthropogenic emissions of NMVOCs drives the enhancement of nitrate formation over eastern China through increasing atmospheric oxidizing capacity remains as the original manuscript. We revised relevant text on P11-12 to reflect the additional results of the NO$_x$ sensitivity test, such as "In addition to NMVOCs, NO$_x$ is another important precursor of O$_3$. We therefore conducted sensitivity simulations to understand the role that foreign anthropogenic emissions of NO$_x$ play in the nitrate enhancement in eastern China. As shown in Fig. 6 (top-bottom) for January, the promotion of nitrate formation is very small with the additional influence of NO$_x$ emitted from foreign sources. This is due to the high local emission of NO$_x$ over eastern China in January that suppresses the O$_3$ formation under a NO$_x$-saturated regime."

(5) "compositions" should be "composition".

**Response: Done.**

(6) Some references are incomplete.

**Response: Fixed Wang et al. 2022 in references.**

**Reference:**

Choi, J., Park, R. J., Lee, H.-M., Lee, S., Jo, D. S., Jeong, J. I., Henze, D. K., Woo, J.-H., Ban, S.-J., Lee, M.-D., Lim, C.-S., Park, M.-K., Shin, H. J., Cho, S., Peterson, D. and Song, C.-K.: Impacts of local vs. trans-boundary emissions from different sectors on PM2.5 exposure in South Korea during the KORUS-AQ campaign, Atmos. Environ., 203, 196–205, doi:https://doi.org/10.1016/j.atmosenv.2019.02.008, 2019.

Jiang, H., Liao, H., Pye, H. O. T., Wu, S., Mickley, L. J., Seinfeld, J. H. and Zhang, X. Y.: Projected effect of 2000-2050 changes in climate and emissions on aerosol levels in China and associated transboundary transport, Atmos. Chem. Phys., 13(16), 7937–7960,

doi:10.5194/acp-13-7937-2013, 2013.

Kurokawa, J. and Ohara, T.: Long-term historical trends in air pollutant emissions in Asia: Regional Emission inventory in ASia (REAS) version 3, Atmos. Chem. Phys., 20(21), 12761–12793, doi:10.5194/acp-20-12761-2020, 2020.

Li, P., Yan, R., Yu, S., Wang, S., Liu, W. and Bao, H.: Reinstate regional transport of PM2.5 as a major cause of severe haze in Beijing, Proc. Natl. Acad. Sci., 112(21), E2739–E2740, doi:10.1073/pnas.1502596112, 2015.

Park, M. E., Song, C. H., Park, R. S., Lee, J., Kim, J., Lee, S., Woo, J.-H., Carmichael, G. R., Eck, T. F., Holben, B. N., Lee, S.-S., Song, C. K. and Hong, Y. D.: New approach to monitor transboundary particulate pollution over Northeast Asia, Atmos. Chem. Phys., 14(2), 659–674, doi:10.5194/acp-14-659-2014, 2014.

Pennington, E. A., Seltzer, K. M., Murphy, B. N., Qin, M., Seinfeld, J. H. and Pye, H. O. T.: Modeling secondary organic aerosol formation from volatile chemical products, Atmos. Chem. Phys., 21(24), 18247–18261, doi:10.5194/acp-21-18247-2021, 2021.

Shrivastava, M., Cappa, C. D., Fan, J., Goldstein, A. H., Guenther, A. B., Jimenez, J. L., Kuang, C., Laskin, A., Martin, S. T., Ng, N. L., Petaja, T., Pierce, J. R., Rasch, P. J., Roldin, P., Seinfeld, J. H., Shilling, J., Smith, J. N., Thornton, J. A., Volkamer, R., Wang, J., Worsnop, D. R., Zaveri, R. A., Zelenyuk, A. and Zhang, Q.: Recent advances in understanding secondary organic aerosol: Implications for global climate forcing, Rev. Geophys., 55(2), 509–559, doi:https://doi.org/10.1002/2016RG000540, 2017.

Wang, S., Li, S., Xing, J., Ding, Y., Hu, S., Liu, S., Qin, Y. and Dong, Z.: Current and future prediction of inter-provincial transport of ambient PM 2 . 5 in China, Atmos. Chem. Phys. Discuss. [online] Available from: https://doi.org/10.5194/acp-2022-368, 2022.

Yan, Y., Zheng, H., Kong, S., Lin, J., Yao, L., Wu, F., Cheng, Y., Niu, Z., Zheng, S., Zeng, X., Yan, Q., Wu, J., Zheng, M., Liu, M., Ni, R., Chen, L., Chen, N., Xu, K., Liu, D., Zhao, D., Zhao, T. and Qi, S.: On the local anthropogenic source diversities and transboundary transport for urban agglomeration ozone mitigation, Atmos. Environ., 245, 118005, doi:https://doi.org/10.1016/j.atmosenv.2020.118005, 2021b.